# Mechanisms of Sodium/Iodide Symporter-Mediated Mammary Gland Iodine Compensation during Lactation

**DOI:** 10.3390/nu14173592

**Published:** 2022-08-31

**Authors:** Min Fu, Yuanpeng Gao, Wenxing Guo, Qi Meng, Qi Jin, Rui Yang, Ying Yang, Yaqi Zhang, Wanqi Zhang

**Affiliations:** 1Department of Nutrition and Food Science, School of Public Health, Tianjin Medical University, Tianjin 300070, China; 2Tianjin Key Laboratory of Environment, Nutrition and Public Health, Center for International Collaborative Research on Environment, Nutrition and Public Health, Tianjin Medical University, Tianjin 300070, China; 3Department of Endocrinology and Metabolism, Tianjin Medical University General Hospital, Tianjin 300070, China

**Keywords:** iodine, lactation, mammary gland, milk, NIS, rat

## Abstract

This research aimed to investigate the compensation mechanism of iodine deficiency and excess in the mammary gland during lactation. Female rats were divided into the low iodine group (LI), the normal iodine group (NI), the 10-fold high iodine group (10HI) and the 50-fold high iodine group (50HI). We measured the iodine levels in the urine, blood, milk, and mammary gland. The protein expression of sodium/iodide symporter (NIS), DPAGT1, and valosin-containing protein (VCP) in the mammary gland was also studied. The 24-hour urinary iodine concentration, serum total iodine concentration, serum non-protein-bound iodine concentration, breast milk iodine concentration, and mammary gland iodine content in the 50HI group were significantly higher than those in the NI group (*p* < 0.05). Compared with the NI group, NIS expression in the 50HI group significantly decreased (*p* < 0.05). DAPGT1 expression was significantly higher in the LI group than in the NI group (*p* < 0.05). The expression level of VCP was significantly increased in the 10HI and 50HI groups. In conclusion, milk iodine concentration is positively correlated with iodine intake, and the lactating mammary gland regulates the glycosylation and degradation of NIS by regulating DPAGT1 and VCP, thus regulating milk iodine level. However, the mammary gland has a limited role in compensating for iodine deficiency and excess.

## 1. Introduction

Breast milk is the only way for exclusively breastfed newborns to obtain iodine. Breastfed infants depend on sufficient maternal iodine intake for optimal growth and neurological development [1,2]. Studies have shown that breast milk iodine concentration (BMIC) typically ranges from 150 to 180 mg/L in countries with sufficient iodine supply [3]. In iodine-deficient areas, BMIC is usually reduced to 50 mg/L [3,4]. Under iodine overexposure conditions, the median BMIC was 240–1000 mg/L, with some women consuming large amounts of seaweed [5]. Currently, data on the reference value ranges of milk iodine is lacking [5,6].

There is a positive correlation between maternal iodine intake and BMIC [7,8], but the data is not uniform [9]. In iodine-sufficient populations, when iodine intake in lactating women is low, there is increased partitioning of iodine into breast milk [10]. The mammary gland may compensate for the iodine nutritional status of breast milk, but the relevant information and data are scarce [11]. 

The iodine transporter sodium/iodide symporter (NIS) is the primary channel for iodine to enter the mammary gland [12,13]. NIS is expressed in the mammary gland on the basolateral membrane of alveolar epithelial cells [5,13]. NIS is usually produced in the endoplasmic reticulum and transferred to the cell membrane via the Golgi complex. A portion of NIS is modified after translation during this process, including phosphorylation and glycosylation [14,15]. The higher the degree of glycosylation is, the more the NIS localized on the cell membrane is and the better the ability of NIS to transport iodine is [16]. Dolichyl-phosphate (UDP-*N*-acetylglucosamine) acetylglucosaminephosphotransferase 1 (DAPGT1) is a glycosylating enzyme involved in the initial step of *N*-linked glycosylation and the main factor regulating NIS glycosylation [17]. Valosin-containing protein (VCP) is involved in the extraction of unfolded or misfolded proteins from the endoplasmic reticulum and can unfold polyubiquitinated proteins to facilitate their degradation by proteasomes [18,19,20]. It has been reported that the overexpression of VCP leads to increased NIS degradation and reduces NIS localization to the cell membrane, thus affecting the ability of NIS to transport iodine [18].

In this study, we fed female Wistar rats with different iodine nutritional statuses. Lactating females were obtained by mating males and females after successful modeling. Then, we investigated changes in the milk iodine levels and the mechanism of NIS-mediated iodine compensation in the mammary gland at different iodine intakes.

## 2. Materials and Methods

### 2.1. Animals and Treatments

A total of 24 female and 20 male Wister rats aged 4 weeks after weaning were purchased from SBF Beijing Biotechnology Co., Ltd. (Beijing, China). This study was approved by the Animal Research Committee of Tianjin Hospital of ITCWM Nankai Hospital (NKYY-DWLL-2021-048). Rats were kept in standard cages at 22 ± 2 °C with a relative humidity of 40–80% under a 12-h light/12-h dark cycle and given free access to food and water. 

Female rats were randomly divided into four groups (*n* = 6): low iodine (LI, 1 μg/d), normal iodine (NI, 6 μg/d), 10-fold high iodine (10HI, 60 μg/d), and 50-fold high iodine (50HI, 300 μg/d). It was calculated as 20 g of diet and 30 mL of water per day for rats. Rats in the LI group were provided with deionized water daily. Other groups were given deionized water supplemented with different amounts potassium iodide. All rats were given a low iodine diet with plenty of other nutrients, and the average dietary iodine content was 50 mg/kg (Trophic Animal Free High-tech Co., Ltd., Nantong, China, TP016ID103). Male rats were fed the same as female rats in the NI group.

After 10 weeks of conditional intervention, female and male rats were cohabited and mated (female/male = 1:1). The day of birth was defined as postnatal day (PND) 0. Rats’ milk and 24-hour (24-h) urine were collected for two consecutive days on PND14 to determine iodine levels. Then, maternal rats were anesthetized with pentobarbital (40 mg/kg, intraperitoneally), and the blood and mammary glands of maternal rats were collected and weighed.

### 2.2. Urine

Maternal rats and their pups were placed in metabolism cages for two days, and 24-h urine from the maternal rats was collected. The urine samples were stored at −80 °C.

### 2.3. Rats’ Milk

Maternal rats were anesthetized with isoflurane and kept anesthetized during milk extraction. Then, the rats were intramuscularly injected with 0.5 mL of veterinary prolactin. Five minutes later, the nipples were rubbed gently and about 0.5 mL of milk was collected using Pasteur straws. Milk was randomly collected three times in two days to determine the iodine concentration in milk.

### 2.4. Measurement of Iodine in the Mammary Gland, Milk, Urine, and Serum

The iodine concentrations in the urine, milk, serum, and mammary gland were detected and analyzed by inductively coupled plasma mass spectrometry ICP-MS (iCAP Q, Thermo Fisher Scientific, Frankfurt am Main, Germany) using Te for mass bias correction.

### 2.5. Thyroid Function Tests

Thyroid-stimulating hormone (TSH), free triiodothyronine (FT3), free thyroxine (FT4), thyroid peroxidase antibody (TPOAb) and thyroglobulin antibody (TgAb) were measured using an automatic Immulite analyzer with a chemiluminescent kit (Sophonix, Beijing, China). Before testing the sample, each kit was controlled with a calibration solution and quality control product. A calibration solution and quality control product (100 μL) were added to each reagent. Samples were tested only after successful calibration and quality control, and each item required 100 μL of serum.

### 2.6. HE Staining

HE staining of the mammary gland was performed with an HE Staining kit (Beyotime, Nantong, China) according to the manufacturer’s instructions. The images (at ×20 magnification) were captured by an Olympus IX81 microscope (Olympus, Tokyo, Japan) in bright-field mode.

### 2.7. Western Blot Analysis

Mammary glands of maternal rats were homogenized in RIPA buffer (20 mM pH 7.5 TRIS-HCl, 150 mM NaCl, 1 mM EDTA, 1% Triton-X100, 0.5% sodium deoxycholate, 1 mM PMSF, and 10 μg/mL leupeptin), incubated on ice for 30 min, and centrifuged at 14,000× *g* for 10 min at 4 °C. Protein concentrations were determined by a bicinchoninic acid (BCA) protein assay kit (Beyo-time). Protein samples were fractionated through SDS-PAGE and transferred onto PVDF membranes (PVDF; Millipore, Billerica, MA, USA). The PVDF membranes were blocked with 5% BSA (Sigma) in 1× Tris-buffered saline Tween for 1 hour at room temperature. Then, the membranes were incubated with primary antibodies of mouse anti-NIS (1:500; Santa Cruz, CA, USA), mouse anti-VCP (1:1000; Bioss, Beijing, China), rabbit anti-DPAGT1 (1:1000; Bioss, Beijing, China), and rabbit anti-GAPDH (loading control, 1:5000; Bioss, Beijing, China) overnight at 4 °C. After washing with TBST, they were incubated with relational horseradish peroxidase (HRP)-conjugated secondary antibodies for 1 h at room temperature. Then, the proteins were detected by chemiluminescence reagents (Sparkjade, Shandong, China) and observed using a ChemiDocTM XRS+ Imaging System (Bio-Rad, Hercules, CA, USA). The protein levels were quantified by densitometry using NIH ImageJ 1.61 software (National Institutes of Health, Bethesda, MD, USA).

### 2.8. Immunohistochemistry

Mammary glands of maternal rats were removed, immediately fixed with 4% paraformaldehyde, and embedded in paraffin. Sections were dewaxed and rehydrated, and the next steps were performed with an ab64261 rabbit-specific HRP/DAB (ABC) IHC Detection Kit (Abcam, Cambridgeshire, UK). Mammary glands were incubated with the primary antibody of rabbit anti-NIS (1:100; Proteintech, Wuhan, China) overnight at 4 °C. Then, the biotinylated secondary antibody was bound to the primary antibody, and the HRP-labeled streptavidin was bound to the secondary antibody. The HRP produced a brown-colored substance at the site of primary antibody binding by reacting with DAB. The images were obtained with an inverted microscope (IX81; Olympus, Tokyo, Japan).

### 2.9. Statistical Analysis

The statistical software package SPSS Statistics version 20.0 (Armonk, NY, USA) and GraphPad Prism v7 (GraphPad Softwre Inc., San Diego, CA, USA) were used for statistical analysis. The variables conforming to a normal distribution were expressed as means ± SD. Comparisons between different groups were performed by a one-way ANOVA, followed by a Student–Newman–Keuls test for multiple comparisons. The variables without normal distributions were expressed as M (P25, P75), and comparisons between groups were performed by a Kruskal–Wallis test. *p* < 0.05 was considered statistically significant.

## 3. Results

### 3.1. Iodine Nutrition and Thyroid Hormone Levels in Lactating Rats

The average serum iodine concentration (SIC) and urinary iodine concentration (UIC) increased with an increase in the iodine intake. The serum total iodine concentration (STIC), serum non-protein-bound iodine concentration (SNBIC), and UIC of the 50HI group were significantly higher than those of the NI group (*p* < 0.05, Table 1).

The TSH levels in the 10HI and 50HI groups fluctuated greatly and showed extremely high values (Figure 1A). As shown in Figure 1B,C, the FT3 and FT4 in the 50HI group were significantly higher than those in the NI group (*p* < 0.05). The levels of FT3, FT4, and TPOAb were stable in the NI group, while the values of the LI, 10HI, and 50HI groups fluctuated greatly (Figure 1B,C,E). The TgAb in the LI group and 50HI group showed maximum values (Figure 1D). The changes in TSH, FT3, FT4, TgAb, and TPOAb indicate that when iodine is deficient or excessive, the thyroid hormone changes greatly, which may lead to hypothyroidism and thyrotoxicosis. However, there is no standard for diagnosing thyroid function in rats.

### 3.2. Effects of Different Iodine Intakes on Breast Milk Iodine Concentration and Mammary Gland Iodine Content in Lactating Rats

To investigate whether maternal iodine intake during lactation affects breast milk iodine concentration (BMIC) and mammary gland iodine content (MIC), we examined iodine in the breast milk and mammary gland. As shown in Table 2, the 50HI group exhibited significantly higher BMIC and MIC compared with the NI group (*p* < 0.05), but no significant difference was found between the other two groups and the NI group (*p* > 0.05). 

### 3.3. Effects of Iodine on Mammary Gland Cell Morphology in Lactating Rats

HE staining was performed to observe changes in the mammary gland during lactation and to explore whether iodine intake affects the morphology of the mammary gland. As shown in Figure 2, the mammary gland was active during lactation. The acini varied in size and were round or oval. The glandular lumen was highly dilated and filled with milk, and the glandular epithelium was compressed and flattened. The mammary gland morphology was similar in different iodine intake groups.

### 3.4. Effect of Iodine on NIS Expression in the Mammary Gland in Lactating Rats

To further explore the mechanism of iodine transport in the mammary gland, we studied the proteins related to iodine transport. The immunohistochemical results showed that NIS was expressed in breast follicular epithelial cells (Figure 3A–D). As shown in Figure 3E,F, compare with the NI group, the expression of glycosylated NIS was increased in the LI group, but decreased in the 10HI and 50HI groups. However, only the 50HI group decreased significantly (*p* < 0.05).

### 3.5. Effect of Iodine on NIS Glycosylation and Degradation in the Mammary Gland during Lactation

The immunohistochemical results showed that DPAGT1 and VCP were expressed in breast follicular epithelial cells (Figure 4A–H). The general protein expressions of VCP and DPAGT1 in the mammary gland are shown in Figure 4I. Compared with the NI group, DPAGT1 expression level was significantly increased in the LI group (Figure 4J). The expression of VCP was higher in the 10HI and 50HI groups when compared with the NI group (Figure 4K).

## 4. Discussion

This study first explored the compensatory effects of the lactating mammary gland on iodine deficiency and excess. NIS is the main iodine transport channel in the mammary gland. The degree of glycosylation and degradation of NIS affects its capacity for iodine transport [16,17,18]. This study is the first to investigate the expression of NIS, DPAGT1 and VCP in the mammary gland under different iodine statuses. 

The WHO proposes that lactating women with suitable iodine status should have a BMIC in the range of 100–200 μg/L [21]. When the BMIC is 150–180 μg/L, it can be considered that the iodine nutritional status of lactation is sufficient [22]. A BMIC < 100 μg/L suggests that lactation may be in an iodine-deficient state [23]. The above studies suggest that the iodine concentration in breast milk is relatively stable, that is, the lactating mammary gland has a strong regulatory ability. However, existing cohort studies of lactating women, which were not published by our team, have found that even lactating women in iodine-appropriate states still have large variations in BMIC (23–4320 μg/L). Azizi F. et al. found a similar phenomenon in iodine-sufficient populations, which is contrary to the common sense of the nutritional community that the mammary gland can stably regulate iodine [24]. Therefore, this study aimed to investigate the compensatory ability of the lactating mammary gland during iodine deficiency and iodine excess. In our study, the BMIC and MIC increased with the increase of iodine intake, but only the 50HI group was significantly higher than the NI group, which may have been caused by insufficient sample size and a large standard deviation or the compensatory effect of the mammary gland on iodine deficiency and excess during lactation. Therefore, we studied the proteins related to iodine transport to further explore the regulation mechanism of mammary gland iodine transport. NIS, a plasma membrane protein classically mediating iodide uptake in the mammary gland, is generally increased in the lactation period, resulting in increased iodine uptake [25]. NIS expression increases under the action of oxytocin in lactation [12,26,27]. Previous studies have shown that with the increase in iodine intake, the mammary gland developed an adaptive response to protect the offspring from excess iodine: NIS protein expression and mRNA levels in the mammary gland decreased [28,29]. As mentioned above, the expression of NIS showed the same trend in our study. Highly glycosylated NIS is more capable of transporting iodine [17]. Sun Y. et al. found, in rat placenta, that the NIS protein transforms from a non-glycosylated to a glycosylated form during iodine deficiency [30]. In the present study, when iodine intake was inadequate, the glycosylated NIS of approximately 60 kDa size was increased. DPAGT1 catalyzes the first step of *N*-linked protein glycosylation [31], and we found that the DPAGT1 expression level was significantly higher in the LI group compared with the NI group. We speculate that DPAGT1 expression increases when iodine is deficient, thereby increasing the degree of NIS glycosylation. The ability of NIS to transport iodine is enhanced, which in turn increases milk iodine concentration. VCP is a hub protein that interacts with more than 30 adaptor proteins involved in various cellular functions [32]. Fletcher A. et al. used mass spectrometry to identify the protein VCP interacting with NIS in breast and thyroid cancer cell models, showing that VCP could specifically bind NIS and directly regulate NIS degradation [18]. In our present study, the expression of VCP was higher in the 10HI and 50HI groups when compared with the NI group. When iodine intake is excessive, VCP expression is increased in the mammary gland, which promotes NIS degradation and reduces the entry of iodine from the blood into the milk.

## 5. Conclusions

Above all, milk iodine concentration is positively correlated with iodine intake. The lactating mammary gland regulates the glycosylation and degradation of NIS by regulating DPAGT1 and VCP, thus compensating for iodine deficiency and excess. However, NIS has a limited ability to regulate the milk iodine level. When iodine is severely excessive, the milk iodine level still fluctuates greatly.

## Figures and Tables

**Figure 1 nutrients-14-03592-f001:**
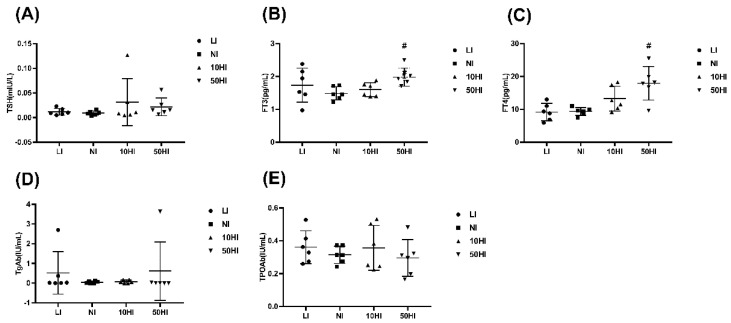
Serum concentrations of TSH (**A**), FT3 (**B**), FT4 (**C**), TgAb (**D**), and TPOAb (**E**) in lactating rats. Free triiodothyronine, FT3; Free thyroxine, FT4; Thyroid−stimulating hormone, TSH; Thyroid peroxidase antibody, TPOAb; Thyroglobulin antibody, TgAb. ^#^
*p* < 0.05 compared with the NI group.

**Figure 2 nutrients-14-03592-f002:**
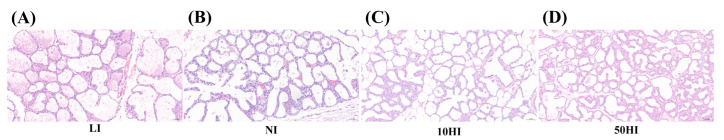
(**A**–**D**) Representative micrographs of HE staining of mammary gland on 14th day of lactation. Scale bars: 50 μm (*n* = 6 per group).

**Figure 3 nutrients-14-03592-f003:**
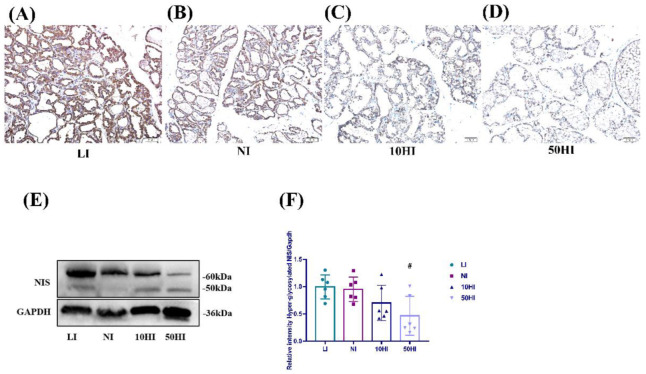
(**A**−**D**) Localization of NIS in the mammary gland on the 14th day of lactation. Scale bars: 50 μm. (**E**) Representative immunoblots of NIS. (**F**) Bar graphs show the semiquantitative levels of Hyperglycosylated NIS determined by band density analysis. (Values are shown as means ± SD. *n* = 6 per group. Comparisons among different groups were performed by one−way ANOVA and followed by an LSD test for multiple comparisons. ^#^
*p* < 0.05 compared with the NI group.)

**Figure 4 nutrients-14-03592-f004:**
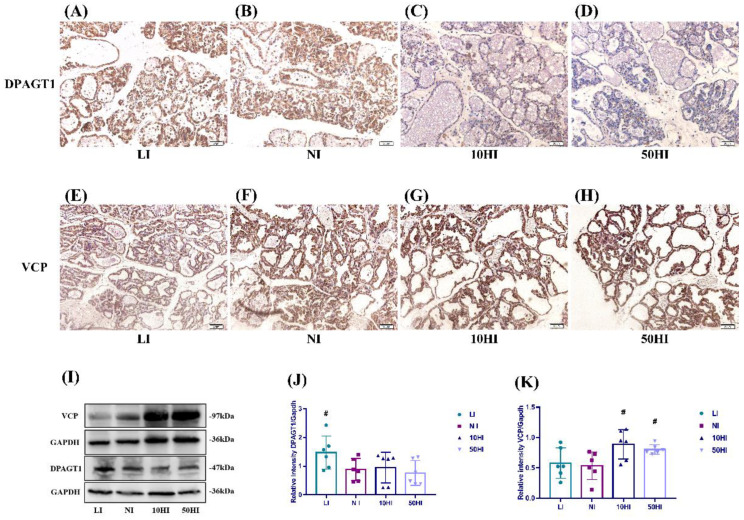
Localization of DPAGT1 (**A**−**D**) and VCP (**E**−**H**) in the mammary gland on the 14th day of lactation. Scale bars: 50 μm. (**I**) Representative immunoblots of DPAGT1 and VCP. Bar graphs show the semiquantitative levels of DPAGT1 (**J**) and VCP (**K**) determined by band density analysis. (Values are shown as means ± SD. *n* = 6 per group. Comparisons among different groups were performed by one−way ANOVA and followed by an LSD test for multiple comparisons. ^#^
*p* < 0.05 compared with the NI group.)

**Table 1 nutrients-14-03592-t001:** Serum iodine and spot urine iodine concentration of lactating rats.

Group	N	Serum		Urine
		STIC (μg/L)	SNBIC (μg/L)	UIC (μg/L) ^a^
LI	6	29.12 (25.08, 33.27)	19.79 (18.61, 21.66)	87.45 (78.71, 95.64)
NI	6	55.77 (42.10, 71.76)	54.33 (41.95, 63.48)	369.96 (312.10, 433.28)
10HI	6	338.18 (218.43, 368.25)	257.19 (220.31, 293.38)	3569.18 (3287.96, 3793.56)
50HI	6	1090.08 (981.04, 1199.83) ^#^	841.83 (758.49, 903.45) ^#^	12,286.04 (11,470.80,13,362.59) ^#^

Serum total iodine concentration, STIC; Serum non-protein-bound iodine concentration, SNBIC; Urinary iodine concentration, UIC. ^#^
*p* < 0.05 compared with the NI group. ^a^ The urinary iodine concentration was the average value over two days.

**Table 2 nutrients-14-03592-t002:** Breast milk iodine concentration and mammary gland iodine content of maternal rats on PND14.

Group	N	BMIC (μg/L) ^a^	MIC (μg)
LI	6	26.46 (22.91, 44.91)	0.38 (0.29, 0.50)
NI	6	220.46 (160.61, 224.36)	0.81 (0.74, 1.37)
10HI	6	1188.75 (936.76, 1473.97)	6.30 (2.37, 8.10)
50HI	6	7413.31 (4226.06, 9175.44) ^#^	72.19 (56.71, 99.63) ^#^

Breast milk iodine concentration, BMIC; Mammary gland iodine content, MIC; Postnatal day, PND. ^#^
*p* < 0.05 compared with the NI group. ^a^ The breast milk iodine concentration was the average value of three random samples.

## Data Availability

The data are contained within the article.

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
