# Peer review of "Mechanisms of Sodium/Iodide Symporter-Mediated Mammary Gland Iodine Compensation during Lactation"

_nutrients, 2022, doi:10.3390/nu14173592_

Round 1
Reviewer 1 Report
This is an important study showing that the breast tissue might compensate and protect at least partially from too high iodine content in breast milk.
The study is well done the results are presented clearly and the discussion is sound
Author Response
Thanks for your suggestions. We checked the paper for language and spelling. Please see the attachment.

Reviewer 2 Report
Authors investigated the mechanism that regulates breast milk iodine concentrations under iodine deficient and excessive conditions using a rat model. They showed changes in protein expression levels in NIS, DPAGT1 and VCP, suggesting the involvement of the sodium/iodide symporter system in the compensatory effects of lactating mammary gland on iodine deficiency and excess.
The reviewer recommends it as a candidate for publication in Nutrients if the following concerns are appropriately addressed.
Major concern:
SD values are considerably large in breast milk iodine concentration in LI group (Table 2) compared to serum iodine concentration in LI group (Table 2), making it difficult to claim the compensatory effects based on the fact that there was no statistically significant difference between LI and NI. It is recommended to increase the size of N (N=10~20 rats).
Minor concern:
In Table 1, # is used in the legend while * is used in Table to show the data with P<0.05. Please correct labels.
Author Response
Dear reviewers,
Thank you very much for giving us an opportunity to improve our manuscript (Mechanisms of Sodium/Iodide Symporter-Mediated Mammary Gland Iodine Compensation During Lactation). We appreciate the editors and reviewers very much for their constructive comments and suggestions on our manuscript. We have modified the manuscript accordingly and marked all the changes in red in the revised manuscript (Please see the attachment), and detailed corrections are listed in ‘Response to Reviewers’ point by point.
Best regards
Comments from the Editors and Reviewers:
Reviewer #2:
- SD values are considerably large in breast milk iodine concentration in LI group (Table 2) compared to serum iodine concentration in LI group (Table 2), making it difficult to claim the compensatory effects based on the fact that there was no statistically significant difference between LI and NI. It is recommended to increase the size of N (N=10~20 rats).
Response: Thanks for your suggestions. Your concern is quite reasonable, and we will increase the sample size in the future study.
Serum iodine concentrations have been relatively stable in many studies[1,2]. We believe that the iodine concentration in serum is the result of compensation by multiple organs, such as the thyroid and kidney. When iodine is deficient, more of the iodine stored in the thyroid gland is released into the serum, and the kidneys expel less iodine. The compensatory ability of breast to iodine can also be determined by the expression level of iodine transporter NIS. In iodine deficiency, the expression of NIS tended to increase. The expression of NIS tended to decrease when iodine was excessive.
- In Table 1, # is used in the legend while * is used in Table to show the data with P<0.05. Please correct labels.
Response: Thanks for your kind remind. We are very sorry for our incorrect writing. The manuscript was checked and modified carefully and all the changes were marked in red in the revised manuscript.
- Zou, Y.; Li, H.; Pang, J.; Liu, X.; Zejipuchi; Tian, L.; Yu, S.; Wang, D.; Hou, L.; Yin, Y.; et al. An evaluation of urine and serum iodine status in the population of Tibet, China: No longer an iodine-deficient region. Nutrition 2021, 82, 111033, doi:10.1016/j.nut.2020.111033.
- Cui, T.; Wang, W.; Chen, W.; Pan, Z.; Gao, S.; Tan, L.; Pearce, E.N.; Zimmermann, M.B.; Shen, J.; Zhang, W. Serum Iodine Is Correlated with Iodine Intake and Thyroid Function in School-Age Children from a Sufficient-to-Excessive Iodine Intake Area. J Nutr 2019, 149, 1012-1018, doi:10.1093/jn/nxy325.

Reviewer 3 Report
This study investigateed the changes in the milk iodine levels and the mechanism of NIS-mediated 64 iodine compensation in the mammary gland at different iodine intakes.
It is thought that additional technologies related to sodium/iodide symporter (NIS), DPAGT1 and Valosin-containing protein (VCP) are required for the research needs.
Discussion on the results of sodium/iodide symporter (NIS), DPAGT1 and Valosin-containing protein (VCP), which are key analysis factors, is considered insufficient.
Author Response
Dear reviewers,
Thank you very much for giving us an opportunity to improve our manuscript (Mechanisms of Sodium/Iodide Symporter-Mediated Mammary Gland Iodine Compensation During Lactation). We appreciate the editors and reviewers very much for their constructive comments and suggestions on our manuscript. We have modified the manuscript accordingly and marked all the changes in red in the revised manuscript (Please see the attachment), and detailed corrections are listed in ‘Response to Reviewers’ point by point.
Best regards
Comments from the Editors and Reviewers:
Reviewer #3:
- It is thought that additional technologies related to sodium/iodide symporter (NIS), DPAGT1 and Valosin-containing protein (VCP) are required for the research needs.
Response: Thanks for your suggestions. We supplemented the results of immunohistochemistry to qualitatively detect the expression of DPAGT1 and VCP in breast follicular cells. In Figure 4A and Figure 4B, we detected VCP and DPAGT1 expression in the cytoplasm of mammary cells. we have added the details in the revised manuscript and marked it in red. (Page 6, lines 206-207)
- Discussion on the results of sodium/iodide symporter (NIS), DPAGT1 and Valosin-containing protein (VCP), which are key analysis factors, is considered insufficient.
Response: Thanks for your suggestions. There are few studies on the effects of DPAGT1 and VCP on the iodine transporter NIS. Current studies have shown that NIS protein transforms from a non-glycosylated to a glycosylated form during iodine deficiency [1]. In the present study, when iodine intake is inadequate, glycosylated NIS of approximately 60 kDa size is increased. DPAGT1 catalyses the first step of N-linked protein glycosylation [2], we found that the DPAGT1 expression level was significantly higher in the LI group compared with the NI group. We speculate that in iodine deficiency, DPAGT1 expression increases thereby increasing the degree of NIS glycosylation, and the ability of NIS to transport iodine is enhanced, which in turn increases milk iodine concentration. VCP is a hub protein that interacts with more than 30 adaptor proteins involved in various cellular functions[3]. Fletcher A et al. used mass spectrometry to identify the protein VCP interacting with NIS in breast and thyroid cancer cell models, showing that VCP could specifically bind NIS and directly regulate NIS degradation[4]. In our present study, the expression of VCP was higher in the 10HI and 50HI groups when compared with the NI group. When iodine intake is excessive, VCP expression is increased in the mammary gland, which promotes NIS degradation and reduces the entry of iodine from the blood into the milk.
The discussion of the results of NIS, DPAGT1 and VCP are added to the discussion section in the revised manuscript and marked in red. (Page 8, lines 252-263)
- Peng, S.; Li, C.; Xie, X.; Zhang, X.; Wang, D.; Lu, X.; Sun, M.; Meng, T.; Wang, S.; Jiang, Y.; et al. Divergence of Iodine and Thyroid Hormones in the Fetal and Maternal Parts of Human-Term Placenta. Biol Trace Elem Res 2020, 195, 27-38, doi:10.1007/s12011-019-01834-z.
- Belaya, K.; Finlayson, S.; Cossins, J.; Liu, W.W.; Maxwell, S.; Palace, J.; Beeson, D. Identification of DPAGT1 as a new gene in which mutations cause a congenital myasthenic syndrome. Annals of the New York Academy of Sciences 2012, 1275, 29-35, doi:10.1111/j.1749-6632.2012.06790.x.
- Jiang, Z.; Kuo, Y.H.; Zhong, M.; Zhang, J.; Zhou, X.X.; Xing, L.; Wells, J.A.; Wang, Y.; Arkin, M.R. Adaptor-Specific Antibody Fragment Inhibitors for the Intracellular Modulation of p97 (VCP) Protein-Protein Interactions. Journal of the American Chemical Society 2022, 144, 13218-13225, doi:10.1021/jacs.2c03665.
- Fletcher, A.; Read, M.L.; Thornton, C.E.M.; Larner, D.P.; Poole, V.L.; Brookes, K.; Nieto, H.R.; Alshahrani, M.; Thompson, R.J.; Lavery, G.G.; et al. Targeting Novel Sodium Iodide Symporter Interactors ADP-Ribosylation Factor 4 and Valosin-Containing Protein Enhances Radioiodine Uptake. Cancer Res 2020, 80, 102-115, doi:10.1158/0008-5472.Can-19-1957.

Round 2
Reviewer 2 Report
The authors have adequately addressed the concerns. The reviewer thinks that the revised manuscript is a good candidate for publication in Nutrients.
Author Response
Thank you very much for your constructive comments and suggestions on our manuscript.
Reviewer 3 Report
The need for research has suggested that additional technologies related to sodium/iodide symporter (NIS), DPAGT1 and Valosin-containing protein (VCP) are required, but this is still insufficient. Others have been modified appropriately.
Author Response
Thank you very much for your constructive comments and suggestions on our manuscript. Due to the negligence of the experimental design, we did not retain excess tissue for further experiments. Your suggestion is very reasonable and we will further improve it in the future study.